# Serum Levels of Urokinase Plasminogen Activator Receptor (suPAR) Discriminate Moderate Uncontrolled from Severe Asthma

**DOI:** 10.3390/jpm12111776

**Published:** 2022-10-28

**Authors:** Ourania S. Kotsiou, Ioannis Pantazopoulos, Georgios Mavrovounis, Konstantinos Marsitopoulos, Konstantinos Tourlakopoulos, Paraskevi Kirgou, Zoe Daniil, Konstantinos I. Gourgoulianis

**Affiliations:** 1Department of Human Pathophysiology, Faculty of Nursing, University of Thessaly, 41110 Larissa, Greece; 2Department of Internal Medicine, University of Thessaly, 41500 Larissa, Greece; 3Department of Respiratory Medicine, University of Thessaly, 41500 Larissa, Greece

**Keywords:** asthma, control, severity, urokinase plasminogen activator receptor

## Abstract

Introduction: The most clinically useful concept in asthma is based on the intensity of treatment required to achieve good asthma control. Biomarkers to guide therapy are needed. Aims: To investigate the role of circulating levels of soluble urokinase plasminogen activator receptor suPAR as a marker for asthma severity. Methods: We recruited patients evaluated at the Asthma Clinic, University of Thessaly, Greece. Asthma severity and control were defined according to the GINA strategy and Asthma Contro Test (ACT). Anthropometrics, spirometry, fractional exhaled nitric oxide (FeNO), suPAR, blood cell count, c-reactive protein (CRP), and analyses of kidney and liver function were obtained. Patients with a history of inflammatory, infectious, or malignant disease or other lung disease, more than 5 pack years of smoking history, or corticosteroid therapy were excluded. Results: We evaluated 74 asthma patients (69% female, mean age 57 ± 17 years, mean body mass index (BMI) 29 ± 6 kg/m^2^). In total, 24%, 13%, 6%, 5%, 29% and 23% of the participants had mild well-controlled, mild uncontrolled, moderate well-controlled, moderate uncontrolled, severe well-controlled, and severe uncontrolled asthma, respectively. Overall, 67% had T2-high asthma, 26% received biologics (15% and 85% received omalizumab and mepolizumab, respectively), and 34% had persistent airway obstruction. suPAR levels were significantly lower in asthmatics with moderate uncontrolled asthma than in patients with severe uncontrolled asthma without (2.1 ± 0.4 vs. 3.3 ± 0.7 ng/mL, *p* = 0.023) or with biologics (2.1 ± 0.4 vs. 3.6 ± 0.8 ng/mL, *p* = 0.029). No correlations were found between suPAR levels and age, BMI, T2 biomarkers, CRP, or spirometric parameters. Conclusions: suPAR levels were higher in asthmatics with severe disease than in those with moderate uncontrolled asthma.

## 1. Introduction

The soluble urokinase plasminogen activator receptor (suPAR) is the circulating form of the cell surface receptor urokinase plasminogen activator receptor (uPAR) (CD87), which is expressed by a plethora of cells ranging from mono- and lymphocytes to endothelial and smooth muscle cells [1]. suPAR is a novel biomarker playing an important role in many physiological and pathological processes, including endothelial dysfunction, thrombosis [2], inflammation [1,2,3,4], chemotaxis [1,2], tissue remodeling [1,2,3,4], and tumorigenesis [1,2]. suPAR’s high stability in plasma samples makes it an ideal candidate biomarker in patients with inflammatory, infectious, and malignant diseases [1,2,3,4,5,6,7]. Recently, a position statement on the prognostic role of suPAR in the screening of patients admitted to the emergency department was issued by the Hellenic Sepsis Study Group.

A remarkable role for suPAR serum levels has been demonstrated in airway diseases. It has been reported that suPAR can be used to evaluate stable chronic obstructive disease (COPD) [3] as a predictor of acute exacerbation and in monitoring response to treatment [3]. Moreover, suPAR has a significant role in increased systemic inflammation associated with coexisting COPD and bronchiectasis [4].

Evidence has shown that airway inflammation might spread into the circulatory system and cause systemic inflammatory injuries. In that context, emerging data support asthma associated with chronic low-grade systemic inflammation, a prothrombotic state, and premature atherosclerosis, even in clinically stable asthma patients [5]. Asthma is also characterized by endothelial dysfunction related to airway obstruction [5]. There is evidence of an abnormal amount of endothelial tissue in asthma and that this tissue and its progenitor cells behave in a dysfunctional manner [6]. Sputum, biopsy and serum suPAR levels were elevated in stable asthma patients compared to controls [7]. Moreover, in asthmatic patients, high suPAR indicated impaired lung function and was shown to correlate with airway resistance [1]. However, no data on the potential role of circulating suPAR as a marker for asthma severity and prognosis according to severity have been reported so far.

Patients with severe asthma are at a particularly high risk of exacerbations, hospitalization and death and have severely impaired quality of life. On the other hand, patients with mild asthma (the silent majority of asthmatics) account for the majority of the morbidity and healthcare resource utilization associated with asthma [1,8].

The aim of this study was to investigate the effectiveness of suPAR as an indicator of the severity of asthma.

## 2. Methods

### 2.1. Study Design

The recruitment period of the study lasted 6 months. Detailed lung function tests were performed in severe asthmatic patients and control groups in the following order: measurement of the fraction of exhaled nitric oxide (FeNO and spirometry) [9]. Severe asthmatics and controls had a set of standard blood tests analyzed, including suPAR, blood cell count (white blood cell count, eosinophils %, absolute eosinophil count), C-reactive protein (CRP), electrolytes (sodium, potassium), and analyses of the kidney (urea, creatinine) and liver function (aspartate transaminase (AST), alanine transaminase (ALT), gamma-glutamyl transpeptidase (γGT), and alkaline phospatase (ALP)) at baseline. The Ethics Committee of the University of Thessaly approved the protocol. Written informed consent was obtained.

### 2.2. Participants

All patients were well-defined regarding asthma severity as they were managed by an experienced pulmonologist in the external Unit of Asthma of the University of Thessaly in Greece for at least one year prior to recruitment. Exclusion criteria were: any history of acute (within four weeks of recruitment) or chronic inflammatory, infectious or malignant disease, COPD and/or other relevant lung diseases causing alternating impairment in lung function, current smoking or more than 5 pack years of smoking history, hypertension, diabetes mellitus, angiopathy, renal disorder, or corticosteroid therapy.

### 2.3. Assessment of the Severity of Asthma

The severity of asthma was assessed according to the level of treatment required to control symptoms and exacerbations according to GINA 2022 [8]. Mild asthma was defined as asthma that was well controlled while treated with as-needed inhaled corticosteroids (ICS)/formoterol or with low-dose ICSs, plus an as-needed short-acting bronchodilator (SABA) [8]. Moderate asthma was defined as asthma that was well-controlled with a low- or medium-dose ICS-long-acting bronchodilator (LABA) (with step 3 or step 4 treatment) [8]. Severe asthma was defined as asthma that required high-dose ICS/LABA to prevent it from becoming uncontrolled or asthma that remained uncontrolled despite this treatment [8].

### 2.4. Asthma Control Evaluation

Asthma control was assessed according to the level of symptom control [8]. Symptom control was determined using the Asthma Control Test (ACT) [10] and discriminated as well-controlled or uncontrolled [8]. In the ACT, scores range from 5–25. Scores of 20–25 were classified as controlled and 5–19 as not well-controlled [10].

### 2.5. Fraction of Exhaled Nitric Oxide (FeNO)

The FeNO (MEDISOFT, MEDICAL GRAPHICS CORP, Minnesota, USA) was performed according to recommendations [11]. Recommended cut-off values for normal FeNO levels were <25 parts per billion (ppb) [11].

### 2.6. Spirometry

Lung function was measured by means of an electronic spirometer (Spirolab FCC ID: TUK-MIR045) according to the American Thoracic Society (ATS) guidelines [12]. Persistent airflow limitation was defined as forced expiratory volume in the first second (FEV1)/forced vital capacity (FVC) ratio consistently < 70% despite irreversibility in asthma expressed as an increase in FEV1 ≥ 12% and 200 mL [8].

### 2.7. Blood Sample Collection

Peripheral venous blood samples were collected in sterile, pro-coagulation tubes and centrifuged immediately; the resulting serum samples were stored at −80 °C until analysis. Plasma suPAR levels were measured using the suPARnostic AUTO Flex ELISA kit (ViroGates A/S, Birkerød, Denmark) as described in detail previously [13]. The suPARnostic ELISA measures the full-length suPAR molecule (D1D2D3) and the cleaved suPAR molecule (D2D3). CRP was measured using a COBAS 6000 analyzer (Roche Diagnostics, Mannheim, Germany).

### 2.8. Statistical Analyses

The Pearson correlation method was used for correlation analysis between pairs of continuous variables. To identify differences between two independent groups, an unpaired t-test was used. Parametric data comparing three or more groups were analyzed with one-way ANOVA and Tukey’s multiple comparisons test, while non-parametric data were analyzed with the Kruskal–Wallis test and Dunn’s multiple comparison test. Pearson’s chi-squared test was used to determine whether there was a statistically significant difference between frequencies. A result was considered statistically significant when the *p*-value was <0.05. Data were analyzed and visualized using SPSS Statistics v. 23 (Armonk, NY, USA, IBM Corp.) and GraphPad Prism 8.

## 3. Results

We evaluated 74 asthma patients. A total of 69% of them were female. The mean age of the population was 57 ± 17 years. The mean body mass index (BMI) was 29 ± 6 kg/m^2^ (Table 1).

Overall, 24%, 13%, 6%, 5%, 29% and 23% of the participants had mild well-controlled, mild uncontrolled, moderate well-controlled, moderate uncontrolled, severe well-controlled, and severe uncontrolled asthma, respectively.

Furthermore, 67% of the population had T2-high asthma according to the measured T2 high biomarkers (blood eosinophil count ≥ 150 cells/µL and FeNO ≥ 20 ppb). Overall, 26% received biologics (15% and 85% received omalizumab and mepolizumab, respectively), while 34% had persistent airway obstruction. The demographic, clinical and spirometric characteristics of the study population are presented in Table 1. Females had significantly less symptom control than males (Table 1).

Abbreviations: BMI, body mass index; FeNO, fraction of exhaled nitric oxide; FEV1, forced expiratory volume in the first second; FVC, forced vital capacity. No significant differences regarding the measured laboratory parameters (white blood cell count, CRP, sodium, potassium, urea, creatinine, AST, ALT, γGT, and ALP were detected among genders. The comparison of serum suPAR levels among groups of asthmatics is presented in Table 1.

suPAR levels were significantly higher in asthmatics with severe uncontrolled asthma not receiving biologics than in patients with moderate uncontrolled asthma (3.3 ± 0.7 vs. 2.1 ± 0.4 ng/mL, *p* = 0.023) (Figure 1). Moreover, suPAR levels were significantly lower in asthmatics with moderate uncontrolled asthma than in patients with severe uncontrolled asthma without (2.1 ± 0.4 vs. 3.3 ± 0.7 ng/mL, *p* = 0.023) or with biologics (2.1 ± 0.4 vs. 3.6 ± 0.8 ng/mL, *p* = 0.029) (Figure 2).

No correlations were found between suPAR levels and age, BMI, T2 biomarkers, white blood cell count, CRP, electrolytes, parameters of the kidney and liver function, or spirometric parameters (Appendix A). The correlation analysis between suPAR levels and the most important parameters is shown in Figure 3.

## 4. Discussion

This study found that suPAR levels were significantly higher in asthmatics with severe uncontrolled asthma with or without biologics than in patients with moderate uncontrolled asthma. No correlations were found between suPAR levels and age, BMI, T2 biomarkers, white blood cell count, CRP, electrolytes, kidney and liver function parameters, CRP, or spirometric parameters.

Previous studies have shown that suPAR is associated with disease progression and severity in multiple diseases. Thus far, few studies have explored suPAR’s role in asthma outcomes [1]. More specifically, elevated suPAR levels have been associated with hospital all-cause readmission and all-cause mortality in hospitalized patients with a diagnosis of asthma made as soon as they were acutely admitted to the emergency department [8]. Another study found that suPAR concentrations were increased in a small cohort of asthmatics with poor disease control compared to patients with well-controlled asthma [1].

In this study, we found no correlation between age and suPAR levels. Nevertheless, there is evidence that age is a non-modifiable risk factor that correlates with an increase in suPAR levels [14]. A previous study in a population of 182 generally healthy individuals aged 74–89 years found that those aged 24–66 years had higher suPAR levels than younger controls: 3.79 ng/mL (95% CI 3.64–3.96 ng/mL) vs. 3.16 ng/mL (95% CI 2.86–3.45 ng/mL) [14]. These levels increased further with advancing age and were similar in women and men. Aging is associated with systematic cardiac and vascular structure alterations due to immunological responses and natural hormonal changes, resulting in a gradual decline in organ function [14]. However, in our study, we excluded asthmatics with comorbidities associated with low-grade chronic inflammation processes such as diabetes, heart failure, and malignant and inflammatory systemic diseases; this could explain the fact that we did not find any association between age and suPAR levels.

Although obesity is considered a low-grade inflammatory disease, or parainflammation, in this study, we found no association between suPAR levels and BMI. Limited data investigate suPAR as an inflammatory biomarker in obesity. More specifically, Kosecik et al. reported that suPAR has no predictive value for future atherosclerosis in obese children after investigating 136 participants with a median age of 12.05 years [15].

We found no association between suPAR levels and T2-high biomarkers. A previous study documented that in patients acutely admitted with asthma, elevated suPAR concentrations together with blood eosinophil count < 150 cells/μL at the time of hospital admission were associated with both 365-day all-cause readmission and mortality, implying that in asthma, the uPAR pathway associates with non-T2 asthma but is implicated in neutrophils and T1/T17 T-cells that are thought to be part of the pathogenesis of the non-T2 asthma endotypes [14,16,17].

Neutrophils are a primary source of circulating suPAR [18]. Studies report the usefulness of suPAR in predicting severe outcomes in critical illness related to inflammatory and infectious diseases [18], along with CRP and leukocytes. In this study, we found no correlation between suPAR and other inflammatory markers, such as white blood cell count or CRP, given that we excluded patients with significant comorbidities. In the same context, no correlation was detected between suPAR and kidney parameters and liver function parameters. However, there is evidence that suPAR plasma levels were significantly higher in patients with chronic kidney disease (7.9 ± 3.82 ng/mL) than in controls (1.76 ± 0.77 ng/mL, *p* < 0.001) and correlated with disease severity [19]. Similar to its prognostic properties in patients with sepsis, serum suPAR concentrations might serve as an interesting biomarker in cirrhosis and acute liver failure [19].

FEV1 levels are not the only factor taken into account to classify disease severity [16]; they have long been known to be one of the major predictors of mortality among individuals with asthma [16]. However, lung function deficits with magnitudes insufficient to cause clinically manifest functional impairments found in mild asthma are also related to molecular pathways that increase susceptibility to the pulmonary effects of exposures. Furthermore, small airway disease (SAD) is highly prevalent in asthma, even in patients with milder disease. Structural alterations at the peribronchiolar level contribute to the pathogenesis of functional abnormalities observed in patients with asthma [16]. Remodeling can affect small airway wall stiffness, thereby changing their distensibility. Given the clinical impact of SAD, its presence should not be underestimated or overlooked as part of the daily management of patients with asthma. SAD is likely to be directly or indirectly captured by combinations of physiological tests, such as spirometry. Notably, suPAR levels have been previously linked to impaired lung function and airway resistance [1]. This study found no correlation between suPAR levels and airway obstruction. Further research is needed to evaluate any potential correlation between suPAR levels and SAD using more detailed techniques.

In asthma, a panel of several cytokines, chemokines, and granule proteins induce airway inflammation and hyperresponsiveness through enhancing innate and type 2 (T2) or non-T2 immune responses [1,8,16]. Although disease severity-related airway inflammation is found in asthma, new evidence has documented persistent chronic airway inflammation and remodeling in mild asthma, except for those with severe asthma, as defined by the treatment step [8]. Neutrophilic asthma is the lesser-known asthma phenotype and is characterized by severe refractory disease. Airway neutrophilia is associated with asthma severity, acute asthma exacerbation, and airflow limitation. However, neutrophils can also be detected in the airways of mild asthmatics [8,16]. Interestingly, studies suggest that the inflammation reflected in circulating suPAR concentrations in part stems from neutrophil activity [9], commonly considered to be non-T2 inflammation [16]. Accordingly, this study found no correlation between suPAR concentrations and T2 biomarkers such as eosinophils and FeNO.

Our study’s findings should be interpreted within the context of its limitations. As such, when considering absolute numbers, our study’s population is small, in a single center with patients from the same population in terms of geo-ethnicity, limiting our findings’ generalizability. Larger multi-center and multi-nation studies are needed to confirm our results. However, other factors could not have confounded our findings, given that we carefully excluded patients with comorbidities associated with high suPAR levels.

## 5. Conclusions

suPAR levels were higher in asthmatics with severe disease receiving or not receiving biologics than in those with moderate uncontrolled asthma. suPAR’s high stability in plasma samples and its noninvasiveness make it an ideal candidate for the management of asthma and the prediction of worse outcomes. The findings of this study suggested a prognostic value of suPAR that would translate into clinical practice in asthma patients and might predict step-up treatment benefits across the spectrum of asthma severity. Furthermore, the information can be used to develop targeted interventions aimed at the risks of so-called mild asthma. An important practical implication is that suPAR might be a useful addition to existing stratification algorithms for identifying patients that particularly benefit from step-up treatment. This study also indicates that suPAR levels might be effective for clinical use associated with specific clinical features, inflammatory phenotypes of asthma, or impaired lung function.

## Figures and Tables

**Figure 1 jpm-12-01776-f001:**
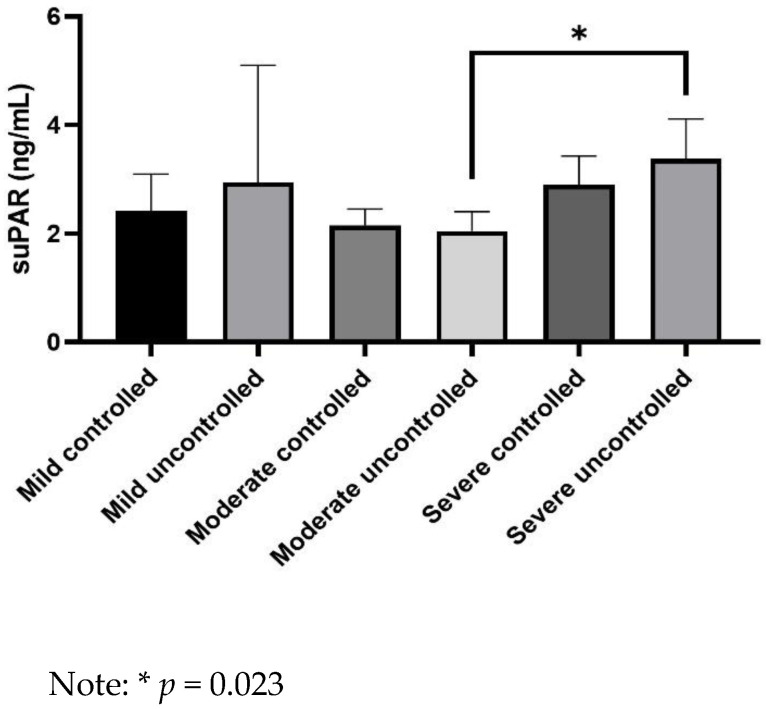
Comparison of serum soluble urokinase plasminogen activator receptor (suPAR) levels between groups of asthmatics.

**Figure 2 jpm-12-01776-f002:**
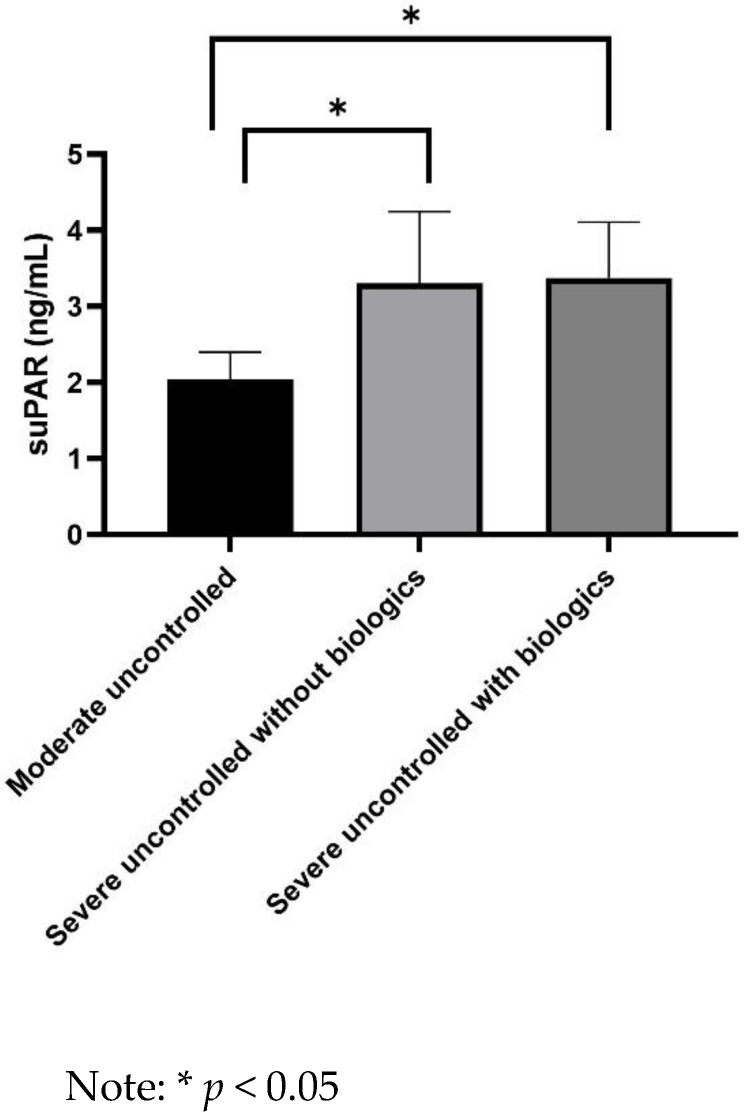
Comparison of serum ssoluble urokinase plasminogen activator receptor (suPAR) levels between moderate uncontrolled and severe uncontrolled asthmatics.

**Figure 3 jpm-12-01776-f003:**
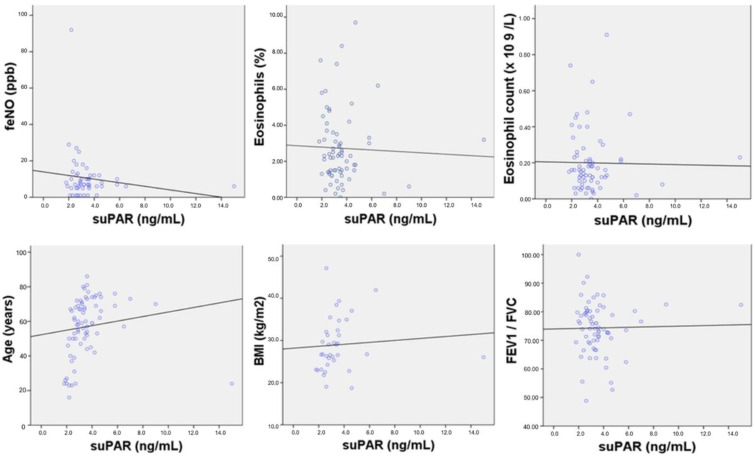
Correlation analysis between suPAR levels and age, body mass index (BMI), T2 biomarkers, and forced expiratory volume in the first second (FEV1)/ forced vital capacity (FVC). Note: No correlation was found between suPAR levels and fractional exhaled nitric oxide (FeNO) (r = −0.014, *p* = 0.308), eosinophils % (r = −0.037, *p* = 0.767), eosinophil count (r = −0.016, *p* = 0.895), age (r = 0.14, *p* = 0.238), BMI (r = 0.079, *p* = 0.646)), and FeV1/FVC (r = 0.020, *p* = 0.870). Abbreviations: BMI, body mass index; FeNO, fractional exhaled nitric oxide; FeV1/FVC, forced expiratory volume in the first second/forced vital capacity.

**Table 1 jpm-12-01776-t001:** Demographic, clinical and spirometric characteristics of the asthmatics (*n* = 74).

Parameter	Study Population (*n* = 75)	Males (*n* = 23)	Females (*n* = 52)	*p*-Value
Age (years)	57 ± 17	52 ± 18	59 ± 16	0.089
BMI (kg/m^2^)	29 ± 6	28 ± 6	29 ± 6	0.758
T2 high phenotype	50 (67)	19 (83)	31 (60)	0.042
Eosinophils (cells per μL)	199 ± 171	194 ± 130	202 ± 185	0.869
FeNO (ppb)	10 ± 3	13 ± 2	9 ± 5	0.216
Mean FEV1/FVC	74 ± 9	73 ± 7	74 ± 9	0.566
Obstruction in spirometry (FEV1/FVC < 70)	25 (34)	7 (30)	18 (34)	0.354
ACT	20 ± 6	22 ± 2	19 ± 5	0.005
Biologics (yes)	19 (26)	6 (26)	13 (25)	0.585

Note: Data are expressed as mean ± SD or as frequencies (percentages).

## Data Availability

The data that support the findings of this study are available on request from the corresponding author, O.S.K.

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
