# Peer review of "Serum Levels of Urokinase Plasminogen Activator Receptor (suPAR) Discriminate Moderate Uncontrolled from Severe Asthma"

_jpm, 2022, doi:10.3390/jpm12111776_

Round 1

Reviewer 1 Report

Generally quite an easy-to-read written piece of work and research. Some comments however as below:

Methodology and results: 

The authors would need to define how control of asthma was graded - based on guidelines ? scoring? spirometry? How did the categories came about? 

Results : T2 high biomarkers - please explain in full. 

How was persistent airway obstruction quantified? a non-reversible bronchodilator test? please explain and mention in full. 

The results are too brief and vague. What were the SuPAR levels for all the categories? Is there a table to show case the different levels and the p values between the different categories? The whole paragraph for results is too simple. 

The discussion also is generally a little too short. Each paragraph should only contain one MAIN idea that the authors would like to highlight from the study, then support it with relevant evidence or current literature. 

The discussion also lacks postulated reasons why there were no correlations found between the parameters. 

Lastly some of the findings could have been confounded by other factors and the authors have failed to highlight these as limitations in a paragraph, eg single centre with patients from the same population in terms of geo-ethnicity etc. How can these findings be generalised to the entire world? Are they valid to be applied in other populations? 

Author Response

Response to Reviewer 1 Comments:

  1. Generally quite an easy-to-read written piece of work and research. Some comments however as below:

RESPONSE: We sincerely thank you for your kind words about our paper. We are delighted to receive positive feedback from you.

  1. Methodology and results: The authors would need to define how control of asthma was graded - based on guidelines ? scoring? spirometry? How did the categories came about? 

RESPONSE: Thank you for the comment. The categories arose according to the asthma severity and symptom control. The severity of asthma was assessed according to the level of treatment required to control symptoms and exacerbations, i.e. mild asthma: treated with as-needed inhaled corticosteroids (ICS)/ formoterol, or with low-dose ICSs, plus as-needed SABA. Moderate asthma: Well-controlled with low or medium dose ICS-LABA (with step 3 or step 4 treatment). Severe asthma: Requires high-dose ICS/LABA to prevent it from becoming uncontrolled, or asthma that remains uncontrolled despite this treatment [pages 2-3, lines 93-102]. Asthma control was assessed according to the level of symptom control as measured by ACT and ACQ; discriminated as well-controlled or uncontrolled [page 3, lines 104-107].

  1. Results : T2 high biomarkers - please explain in full. 

RESPONSE: Thank you for this remark. We explained T2 high biomarkers on page 4, line 145.

  1. How was persistent airway obstruction quantified? a non-reversible bronchodilator test? please explain and mention in full. 

RESPONSE: Thank you for this point. In the revised manuscript, we explained the definition of persistent airflow limitation (page 3, lines 114-117).

  1. The results are too brief and vague. What were the SuPAR levels for all the categories? Is there a table to show case the different levels and the p values between the different categories? The whole paragraph for results is too simple. 

RESPONSE: Thank you for the comment. In the revised manuscript we have added Figure 1 to show the different levels of suPAR levels between the different categories and enriched the results section.

  1. The discussion also is generally a little too short. Each paragraph should only contain one MAIN idea that the authors would like to highlight from the study, then support it with relevant evidence or current literature. 

RESPONSE: Thank you for the comment. The discussion was revised accordingly.

  1. The discussion also lacks postulated reasons why there were no correlations found between the parameters. 

RESPONSE: Thank you for the comment. The discussion was revised postulating reasons why there were no correlations found between the parameters [pages 6-7. lines 190-265].

  1. Lastly some of the findings could have been confounded by other factors and the authors have failed to highlight these as limitations in a paragraph, eg single centre with patients from the same population in terms of geo-ethnicity How can these findings be generalised to the entire world? Are they valid to be applied in other populations? 

 RESPONSE: Thank you for this comment. We have now revised accordingly by adding a paragraph describing the limitations of the study.

We appreciate you taking the time to offer us your insights related to the paper. We found your feedback very constructive. We tried to be responsive to your concerns.

Reviewer 2 Report

Manuscript reference: jpm-1954642

Type of article: brief report

Title: Soluble Urokinase Plasminogen Activator Receptor (suPAR) 2 Serum Levels Discriminate Moderate Uncontrolled From Severe Asthma.

The idea of the study is not bad that suPAr could be indicator of severe asthma, while the statistical methods here are lacking, the description of the methods is also parsimonious. The results are presented in a single table. The results of the correlation are also not shown.

I have several remarks:

- The title. It is not reads well. I suggest changing the title to” Serum Levels of Urokinase Plasminogen Activator Receptor (suPAR) 2 Discriminates Moderate Uncontrolled From Severe Asthma.

- Line 58; “ There is evidence that there is a..” Please rearrange the sentence, too much there is.

- Methods; lack of information on statistics applied. What kind of tests did you use?

- Results, Table 1;  How did you define Th2 high phenotype, please include some explanation in the text.

- Results, Line 125; when you write that there is no correlation, I would like to see some graphs showing that.

- Line 138, there is something wrong with this fragment “cohort of outpatient asthma patients”. Please correct.

- Why in the conclusion the authors write in the future tense; “study will demonstrate, this study will provide”. I think it has already shown and delivered. Another question is whether such a single study is sufficient to state something like this categorically. I would use the words indicates, suggests.

Author Response

Response to Reviewer 2 Comments:

  1. The idea of the study is not bad that suPAr could be indicator of severe asthma, while the statistical methods here are lacking, the description of the methods is also parsimonious. The results are presented in a single table. The results of the correlation are also not shown.

RESPONSE: Thank you for the comments. In the revised manuscript, we have now provided the statistical methods, and more detailed results.

  1. I have several remarks: The title. It is not reads well. I suggest changing the title to” Serum Levels of Urokinase Plasminogen Activator Receptor (suPAR) 2 Discriminates Moderate Uncontrolled From Severe Asthma.

RESPONSE: Thank you for this comment. The title has been changed, as suggested.

  1. Line 58; “There is evidence that there is a..” Please rearrange the sentence, too much there is.

RESPONSE: Thank you for this point. The sentence has been revised.

  1. Methods: lack of information on statistics applied. What kind of tests did you use?

RESPONSE: Thank you for this comment. In the revised manuscript, we provide the statistical analyses that have been performed (page 3, lines 128-137).

  1. Results, Table 1; How did you define Th2 high phenotype, please include some explanation in the text.

RESPONSE: Thank you for this remark. We explain T2 high biomarkers on page 4, line 145.

  1. Results, Line 125; when you write that there is no correlation, I would like to see some graphs showing that.

RESPONSE: Thank you for the comment. In the revised manuscript, we provided Figure 3 and Supplementary Table 1, showing the correlation analyses between suPAR levels and various parameters.

  1. Line 138, there is something wrong with this fragment “cohort of outpatient asthma patients”. Please correct.

RESPONSE: Thank you for the comment. The sentence has been revised.

  1. Why in the conclusion the authors write in the future tense; “study will demonstrate, this study will provide”. I think it has already shown and delivered.

RESPONSE: Thank you for the remark. The sentences have been revised accordingly.

  1. Another question is whether such a single study is sufficient to state something like this categorically. I would use the words indicates, suggests.

RESPONSE: RESPONSE: Thank you for the remark. The sentences have been revised accordingly.

We appreciate you taking the time to offer us your insights related to the paper. We found your feedback very constructive. We tried to be responsive to your concerns.

Round 2

Reviewer 1 Report

No further comments 

Reviewer 2 Report

The article has been improved according to my suggestion, therefore I have no further remarks.